# Human skeletal muscle plasmalemma alters its structure to change its $Ca^{2+}$-handling following heavy-load resistance exercise

Tanya R. Cully[1], Robyn M. Murphy[2], Llion Roberts[3,4], Truls Raastad[5], Robert G. Fassett[3], Jeff S. Coombes[3], Izzy Jayasinghe[1,6] & Bradley S. Launikonis[1]

High-force eccentric exercise results in sustained increases in cytoplasmic $Ca^{2+}$ levels ($[Ca^{2+}]_{cyto}$), which can cause damage to the muscle. Here we report that a heavy-load strength training bout greatly alters the structure of the membrane network inside the fibres, the tubular (t-) system, causing the loss of its predominantly transverse organization and an increase in vacuolation of its longitudinal tubules across adjacent sarcomeres. The transverse tubules and vacuoles displayed distinct $Ca^{2+}$-handling properties. Both t-system components could take up $Ca^{2+}$ from the cytoplasm but only transverse tubules supported store-operated $Ca^{2+}$ entry. The retention of significant amounts of $Ca^{2+}$ within vacuoles provides an effective mechanism to reduce the total content of $Ca^{2+}$ within the fibre cytoplasm. We propose this ability can reduce or limit resistance exercise-induced, $Ca^{2+}$-dependent damage to the fibre by the reduction of $[Ca^{2+}]_{cyto}$ to help maintain fibre viability during the period associated with delayed onset muscle soreness.

[1] School of Biomedical Sciences, The University of Queensland, Brisbane, Queensland 4072, Australia. [2] Department of Biochemistry & Genetics, La Trobe Institute for Molecular Science, La Trobe University, Melbourne, Victoria 3086, Australia. [3] School of Human Movement and Nutritional Sciences, The University of Queensland, Brisbane, Queensland 4072, Australia. [4] Centre of Excellence for Applied Sport Science Research, Queensland Academy of Sport, Brisbane, Queensland 4111, Australia. [5] Norwegian School of Sport Sciences, Oslo N-0806, Norway. [6] School of Biomedical Sciences, University of Leeds, Leeds LS2 9JT, UK. Correspondence and requests for materials should be addressed to B.S.L. (email: b.launikonis@uq.edu.au).

Demanding bouts of running or resistance exercise are known to have long-lasting consequences for the internal environment of the muscle fibre. These types of exercise involve eccentric contractions, where the muscles lengthen while tension is developed. An eccentric workload can cause muscle damage and induce soreness in the days following exercise, commonly referred to as delayed onset muscle soreness (DOMS). The type of damage observed is structural damage to sarcomeres, increased permeability of the plasmalemma and reduced efficiency of the $Ca^{2+}$ release apparatus[1].

A major contributor to the damage seen in muscle fibres following eccentric contractions is due to $Ca^{2+}$ entry into the muscle, which increases the basal level of cytoplasmic $[Ca^{2+}]$ ($[Ca^{2+}]_{cyto}$) to activate calpains[2–4]. $Ca^{2+}$ may enter the muscle through non-specific pathways in the permeant plasmalemma, an event that occurs presumably post-exercise. During exercise, $Ca^{2+}$ entry is excitation-dependent. Gissel and Clausen[5,6] have shown increases in muscle calcium content in response to muscle activity; and $Ca^{2+}$ imaging experiments have confirmed that there is an action potential-activated $Ca^{2+}$ current, which is tightly associated with individual action potentials[7].

In human muscle, eccentric contraction causes a significant increase in the muscle calcium content, depending upon the exercise and the duration of the exercise[8–10]. Interestingly, in the muscle stressed by exercise involving eccentric contractions, damage can be absent from the majority of the fibres exposed to the insult[11,12]. This result is suggestive that the muscle employs a protective mechanism to maintain fibre viability while it recovers from the bout of demanding exercise.

A unique feature of the muscle post-eccentric contractions is the appearance of persistent vacuoles. Such structures do not form following a similar workload consisting of only concentric contractions[13]. These vacuoles are localized and do not align with the sarcomeric inhomogeneities caused by the eccentric contractions[13,14]. Vacuoles form within the tubular (t-) system, which is a network of tubules that invaginate from the plasmalemma to reach every sarcomere of the fibre[15]. The t-system network is comprised of transverse tubules and longitudinal tubules[16,17]. Both tubule types have distinct functional roles. The transverse tubules support excitation-contraction coupling by housing voltage-sensitive molecules that directly activate the sarcoplasmic reticulum (SR) ryanodine receptor (RyR) to release $Ca^{2+}$ in response to action potentials to raise $[Ca^{2+}]_{cyto}$ several-fold. Transverse tubules also exchange $Ca^{2+}$ with the cytoplasm via $Na^{+}$–$Ca^{2+}$ exchangers (NCX) and the plasma membrane CaATPase (PMCA) to support $Ca^{2+}$ uptake from the cytoplasm[18]; and transverse tubular Orai1 (ref. 19) coupled to SR STIM1L[20] support store-operated $Ca^{2+}$ entry (SOCE; refs 21,22). Longitudinal tubules support the spread of excitation across the muscle[23,24].

The source of the vacuoles within the t-system is specifically the longitudinal tubules, which become sinks that sequester small molecules from the transverse tubules across a tight luminal junction that exclude the entry of large molecules[16]. The ability of the t-system to increase its volume and sequester small molecules in response to eccentric contractions[13] grants it the potential to sequester and hold large amounts of calcium. The sequestered $Ca^{2+}$ would effectively be quarantined and prevented from initiating damage at sites within the cytoplasm of the fibre[2,3]. However, it is not known whether vacuoles form in the t-system of human skeletal muscle fibres post-eccentric exercise, or whether their onset and decline parallels that of DOMS. Furthermore, a hypothesis that vacuoles protect the muscle post-eccentric exercise from extensive $Ca^{2+}$-induced damage requires a description of the $Ca^{2+}$-handling properties of the vacuoles, which is currently lacking. To do this would require the spatial discrimination of the $Ca^{2+}$-handling properties of vacuoles from the transverse tubules as these structures sit in their natural position in the fibre, as reductionist approaches such as isolation of vacuoles from the muscle would likely cause them to collapse, as they rely on intrinsic hydrostatic pressure[25].

To determine whether vacuolization of the t-system provides a 'safety net' that sequesters calcium while the muscle recovers from a bout of eccentric contractions, we employed recently developed techniques to reconstruct the three-dimensional (3D) structure of t-system network[17] and to spatially resolve the $Ca^{2+}$-handling properties of the t-system[18] in skeletal muscle fibres isolated from needle biopsies, taken from human subjects before and after exercise. This methodological approach allowed us to track sub-micron-scale changes in both the structure and the $Ca^{2+}$-handling properties throughout large sections of the muscle fibres at an unprecedented level of spatial detail. The exercise protocol was normal heavy-load strength training of leg muscles, involving eccentric (lengthening) muscle actions, which produce muscle damage and DOMS. Muscle soreness and increases in blood creatine kinase activity in the days post-exercise was reported[26]. From these biopsies we observed the t-system network to transiently vacuolate post-exercise, in a process lasting at least 2 days. This change allowed the t-system network to sequester $Ca^{2+}$ from the cytoplasm and therefore lower the calcium content of the muscle during the period of time associated with DOMS[27].

## Results

In this section we examine human skeletal muscle fibres isolated from needle biopsies of the mid *vastus lateralis* to describe the structure of the t-system before and after heavy-load resistance exercise. We also determine the $Ca^{2+}$-handling properties of the t-system in human skeletal muscle fibres, delineating the properties of the transverse tubules and vacuoles. It is accepted that a muscle biopsy from a small region of the large *v. lateralis* muscle is representative of that muscle for biochemical and physiological determinations (although not for determining fibre type distribution[28]). Our approach allows us to sample multiple fibres from each biopsy, so we could have many repeat measures from the same biopsy from a given time point or imposed ionic conditions.

**The t-system structure of muscle before and after exercise**. The 3D structure of the human muscle t-system reconstructed from serial confocal images[17] of isolated fibres from biopsies obtained from subjects prior to exercise is shown (Fig. 1). A shallow (5-μm deep) sub-volume of a fibre (Fig 1a) that was reconstructed in full is colour coded showing the transverse tubules (grey) and longitudinal tubules (red). Note that there are two transverse tubules per sarcomere that approximately localize with the junctions of the A-I bands. See Supplementary Fig. 1A–D for full reconstruction. At close inspection of the reconstruction, regions abundant of longitudinal tubules were identified; often these extended as a series of longitudinal tubule networks across multiple sarcomeres (Fig. 1b), typically in between myofibrils that lack alignment between sarcomeres (Supplementary Fig. 1E,F). The transverse elements by comparison were far more extensive, observed to be encircling the myofibrillar spaces in the xz (transverse) view of a 2-μm deep projection of the fibre yielding to an extensive and highly interconnected network across the thickness of the fibre (Fig. 1c). A 45-μm deep projection in a similar view showing only the longitudinal networks illustrates their non-uniform distribution, highly localized near the surface (periphery) of the fibre (Fig. 1d).

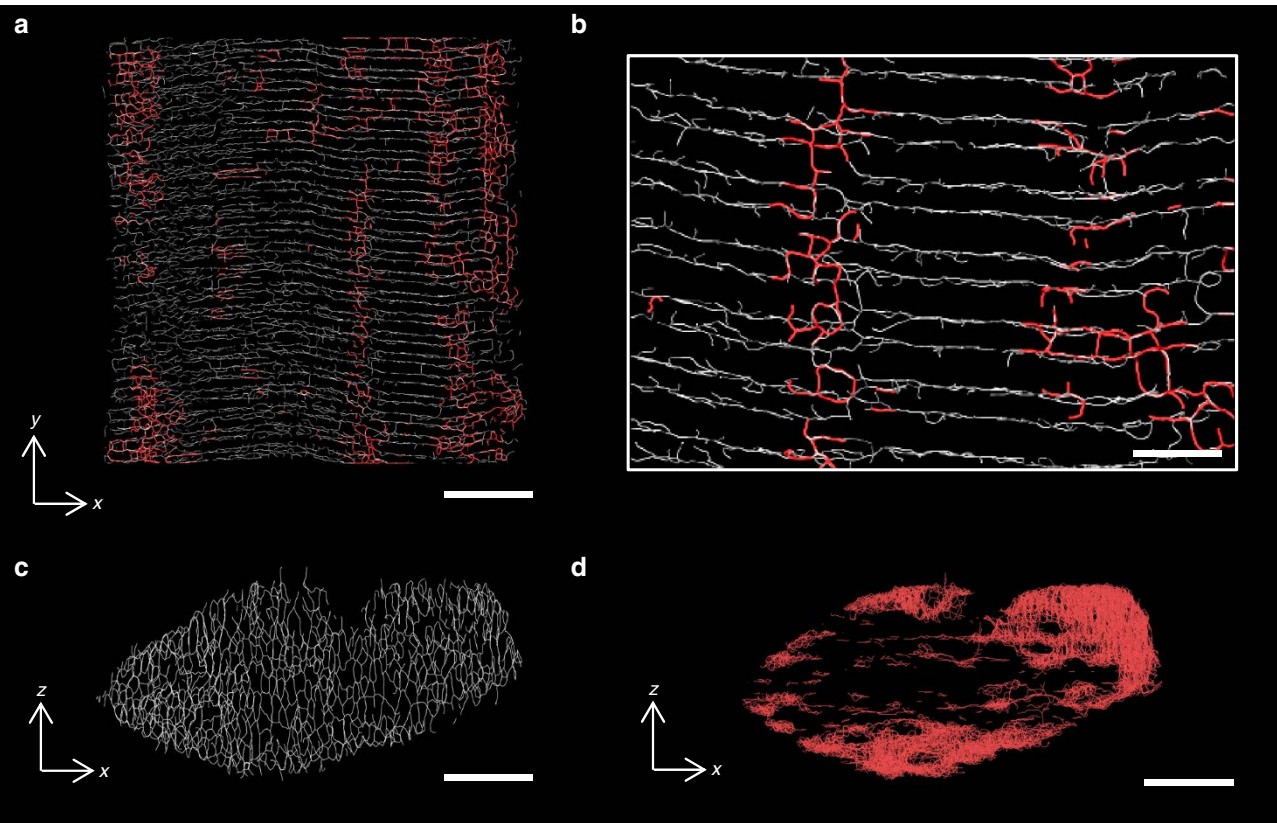

**Figure 1 | 3D structure of the human skeletal muscle t-system.** (**a**) The longitudinal (*xy*) view of 5-μm deep slice of the 3D skeleton of the t-system imaged in a confocal z-stack illustrates the predominately transverse tubule connections (coloured white) making up the network. Longitudinally connecting tubules are coloured in red. (**b**) Magnified view of the transversely sectioned skeleton is shown with the longitudinally connecting tubules in red. Notably, the longitudinal tubules appear to run in series, spanning multiple sarcomeres and containing elements of varying orientation. (**c**) Transverse (*xz*) view of thin, 2-μm deep slice illustrates the intricate network formed by the transverse tubules. (**d**) A full transverse (*xz*) projection of only the subset of longitudinal tubule (red) networks within the reconstructed volume (45 μm long in y-dimension) illustrates that the longitudinal tubules are more likely to be located near the periphery of the fibre and less so in the centre. Scale bars: **a,c,d**: 10 μm, **b**: 2 μm.

Single confocal planes from deconvolved volume images of the t-system from an individual before, 24 and 48 h post an acute strength training session are shown in Fig. 2a. The most notable change in the t-system structure before and after training is the highly fluorescent longitudinal series of vacuoles. These vacuoles were large ( $\varnothing$ 0.8–1 μm, which is well above the resolution limit of the confocal microscopy technique; Supplementary Fig. 2) and approximately tenfold larger than the mean width that we estimate in the non-vacuolated t-system (Supplementary Fig. 3). The characteristic locations and frequencies of these series of vacuoles (for example, Fig. 2a) suggest that they arise from the series of longitudinal tubules observed in the healthy fibres. Notably, vacuoles formed in the post-exercise t-system (24 h) were still present 48 h after exercise, but were diminished in the biopsies obtained 6 days following cessation of strength training (Supplementary Fig. 4). We also observed vacuolation of the sub-sarcolemmal t-system network (Supplementary Fig. 5), which is a lattice of tubules between the two outermost myofibrils of the fibre[29], captured in our confocal 3D z-stacks spanning the full depth of the fibre. Analysis of a total of 24 fibres is included in Fig. 2. This analysis includes fibres from muscle biopsies obtained before exercise (11 image sets from different regions of 5 fibres), 24 h post exercise (14 image sets from 9 fibres) and 48 h post exercise (11 image sets from 10 fibres) showed that the fibre volume occupied by vacuoles increased seven to ninefold following exercise (Fig. 2b). This volume was sustained in the fibres from the biopsies taken 48 h following exercise; however,

the number of vacuoles continued to increase (Fig. 2c) while there was a significant increase in the roundness of the vacuoles between 24 and 48 h (Fig. 2d; all statistics reported in figure legend). We note that regardless of fibre type, we observed vacuolation of the t-system post-eccentric exercise. This is consistent with observations of vacuoles in mouse fast- and slow-twitch fibres[13,17,30]. Furthermore, two of the three subjects in this section of the study was subject to cold water immersion for 10 min following the exercise protocol[26]. Vacuole formation and maintenance requires the action of the t-system $Na^+$ pump[13,25] and is therefore temperature-dependent. No observable difference in vacuole formation could be resolved, nor was expected, at the temporal resolution of our measurements (24 h) in regard to a 10-min exposure to cold water because normal leg muscle temperature was maintained for the vast majority of the time across the sampling period (48 h) following exercise. For this reason the results collected from all biopsies have been pooled.

Another notable change observed is the loss of the regularity of the transverse elements of the t-system (Fig. 2a,e). However, there was regional heterogeneity observed with this change. The fractional histogram of the tubule directionality in Fig. 2e illustrates the predominantly transverse tubule angles ( $\sim 0°$ ) observed in the pre-training biopsies (red plot and left inset image) are diminished in some of the regions exhibiting this disorganized tubule structure (green plot, right inset).

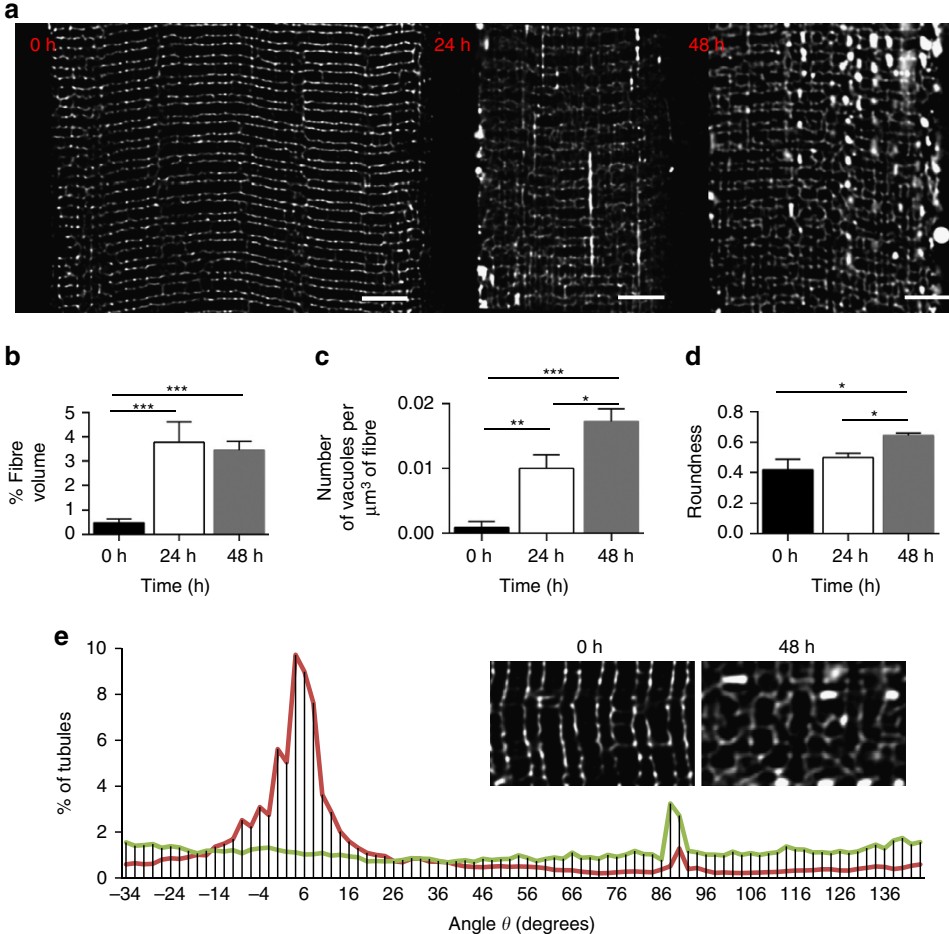

**Figure 2 | Remodelling of the fine structure of the t-system following eccentric exercise.** (**a**) Deconvolved grey scale confocal optical sections of muscle fibres from biopsies prior to (0 h) exercise and 24 and 48 h following exercise illustrate the notable increase in brightly fluorescent vacuoles in the t-system. Notably, these vacuoles appeared in longitudinal series, many of which spanned multiple sarcomeres. (**b**) Analysis of the vacuole in these 3D confocal volumes reported a > threefold increase in the percentage of the fibre volume, which consisted of vacuoles within the first 24 h following exercise. No statistically significant change in this fraction was seen in the subsequent 24 h. (**c**) The density of vacuoles per unit fibre volume was increased 48 h following exercise. (**d**) Vacuole roundness, calculated the inverse of the aspect ratio, was significantly increased at 48 h following exercise. (**e**) Deviation from the predominantly transverse tubule orientation was observed in some regions of the biopsy fibres 48 h following exercise (inset). A histogram of the percentage of t-system elements plotted as a function of the local direction ($\sim 0°$ at the transverse plane and 90° in parallel with the fibre's longitudinal axis) revealed that remodelled regions of fibres sampled 48 h after exercise lacked dominating fraction of transverse tubules; rather a near-random directionality distribution was seen. Scale bars: 5 μm. Tukey's multiple comparison test *P*-values for **b**,**d**: \*, \*\* and \*\*\* refer to < 0.05, < 0.005 and 0.0001, respectively. Data in **b**–**c** presented as mean ± s.e.m.

**$Ca^{2+}$-handling by the transverse tubules and vacuoles.** To test one of the primary functional ($Ca^{2+}$-handling) properties of the transverse tubules and vacuoles of the t-system of human skeletal muscle, we conducted a series of multi-compartment fast $Ca^{2+}$ tracking experiments with fluorescence optical sectioning. For this, resting muscle biopsies were obtained from healthy, recreationally active men and women ($n = 7$). Rhod-5 N trapped in the t-system of mechanically skinned fibres from biopsies was continuously imaged in xyt mode during changes in internal bathing solutions that induced a unidirectional flux of $Ca^{2+}$ across the t-system membrane. A unidirectional flux was generated by either depleting the SR of $Ca^{2+}$ with caffeine (to induce SOCE) or applying a known $[Ca^{2+}]_{cyto}$ under otherwise resting ionic conditions (to induce $Ca^{2+}$ uptake by the $Ca^{2+}$-depleted t-system). The t-system rhod-5N signal ($t$) was converted to $[Ca^{2+}]_{t-sys}$ ($t$) using a recently established method[18]. An example of $[Ca^{2+}]_{t-sys}$ ($t$) during application of caffeine or $[Ca^{2+}]_{cyto}$ is displayed in Fig. 3a. Images of the $[Ca^{2+}]_{t-sys}$ corresponding to points within the transient in a are shown in b. The spatial resolution of the image is

significantly lower than the images in Figs 1 and 2. This was done to prevent bleaching of the t-system trapped dye during continuous imaging in these experiments[18]. In these images there is no indication of vacuolation but a clear transverse striated pattern in the presence of high $[Ca^{2+}]_{t-sys}$. From the seven biopsies, a total of 27 fibres were analysed and for each fibre displaying a regular t-system (largely transverse tubules; Fig. 1), the steady state $[Ca^{2+}]_{t-sys}$, peak t-system $Ca^{2+}$ uptake flux, peak store-dependent $Ca^{2+}$ flux and myosin heavy chain isoform were determined (Fig. 3c–e; refs 18,31). Increasing $[Ca^{2+}]_{cyto}$ was found to increase the steady state $[Ca^{2+}]_{t-sys}$ and the peak uptake flux of the t-system (Fig. 3c,d). Store-dependent $Ca^{2+}$ influx was dependent on the $[Ca^{2+}]_{t-sys}$, but not directly proportional, as is the case in rat muscle[18,32]. A regression line with a gradient of $-0.42$ fitted the data points (Fig. 3e), suggesting that there may be inhibitory regulation of either STIM1L or Orai1 in human muscle. Three fibre types (type I, type IIa and hybrid type I/IIa) were identified in fibres obtained from biopsies used for the analysis presented in Fig. 3.

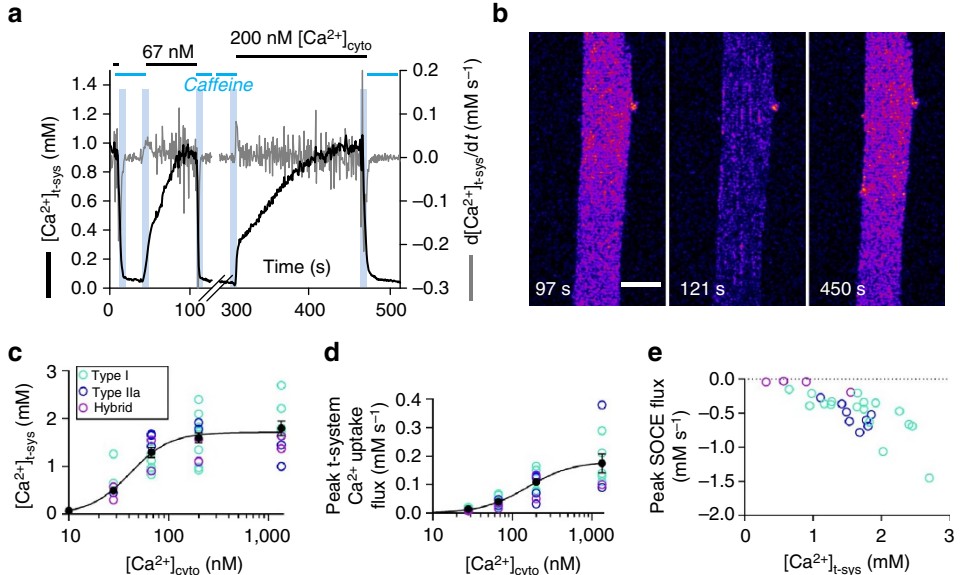

**Figure 3 | $Ca^{2+}$-handling by the human muscle t-system.** (**a**) $[Ca^{2+}]_{t\text{-sys}}$ ($t$) in a human fibre after exposure to caffeine, 67 nM free $[Ca^{2+}]_{cyto}$ and 200 nM $[Ca^{2+}]_{cyto}$. t-system $Ca^{2+}$ flux was derived from $[Ca^{2+}]_{t\text{-sys}}$ ($t$)[18]. The vertical pale blue bars indicate the timing of solution changes. The internal solution is indicated above the horizontal bar. (**b**) confocal image of t-system-trapped $Ca^{2+}$-sensitive dye in a type I fibre from female, 45 years old. (**c**) the average steady-state $[Ca^{2+}]_{t\text{-sys}}$ at known $[Ca^{2+}]_{cyto}$. (**d**) peak t-system $Ca^{2+}$ uptake at known $[Ca^{2+}]_{cyto}$. (**e**) peak SOCE flux at known $[Ca^{2+}]_{t\text{-sys}}$. A regression line of $-0.42 \times [Ca^{2+}]_{t\text{-sys}} + 0.19$ could be fitted to all the data points (not drawn). Note (**c–e**) are results from seven individuals (females, 38 and 45 years old; males, 18, 22, 22, 26 and 42 years old). The coloured circles in (**c–e**) represent type I, IIa and hybrid I/IIa fibre types as indicated. In **c,d**, the mean ± s.e.m. is overlaid in black. The number of values, fibres and $r^2$ values for the fitted curves are: 47, 43 and 29; 36, 30 and 21; 0.785, 0.817 and 0.602, for **c,d,e**, respectively. Scale bar: 40 μm.

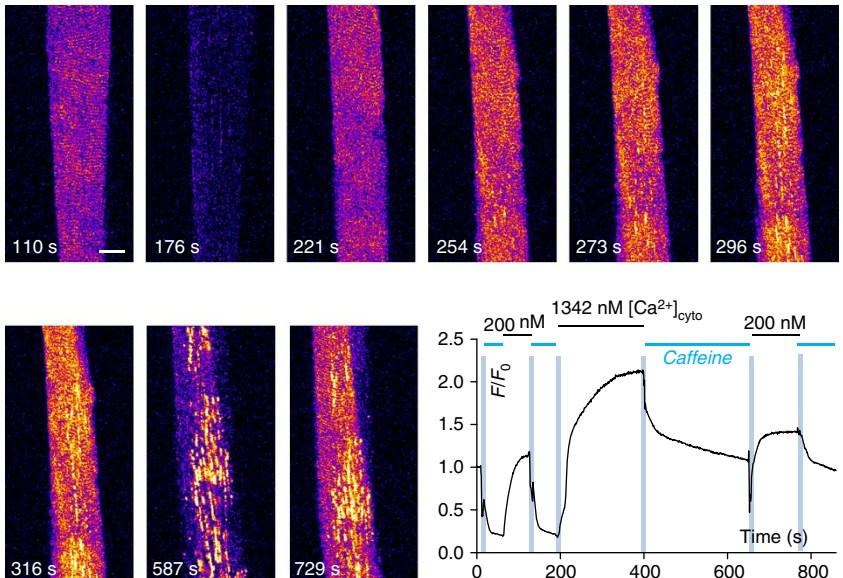

**Figure 4 | Acute formation of vacuoles and vacuole resistance to SOCE.** An *xyt* series of confocal images of t-system rhod-5N shows the t-system responding to caffeine-induced SOCE and reuptake of $Ca^{2+}$ (images 110, 176 and 221 s). The spatially averaged profile at bottom right of the figure indicates the internal bathing solution the fibre was immersed in. The pale blue vertical bars on the graph indicate the timing of the solution changes and the horizontal bars at top indicate the internal solution composition. The time-stamp on the images correspond to that on the graph. The initial exposure of the fibre to caffeine caused SR $Ca^{2+}$ depletion and SOCE that caused a uniform depletion of $[Ca^{2+}]_{t\text{-sys}}$. The following substitution of caffeine for 1.3 μM $[Ca^{2+}]_{cyto}$ caused the t-system to take up $Ca^{2+}$ and for vacuoles to form (longitudinal structures, seen as bright yellow in images 254 to 729 s). The substitution of 1.3 μM $[Ca^{2+}]_{cyto}$ for caffeine caused the transverse tubules to deplete of $Ca^{2+}$ without affecting the $Ca^{2+}$ held in the vacuoles (bright longitudinal structures remain in the presence of caffeine and the spatially averaged fluorescence signal indicated in the graph does not drop as low as in the initial exposure to caffeine). The secondary exposure to 200 nM $Ca^{2+}$ allows the transverse tubules to again take up $Ca^{2+}$ and the final exposure to caffeine again causes only a partial reduction in the spatially averaged $[Ca^{2+}]_{t\text{-sys}}$. Scale bar: 25 μm.

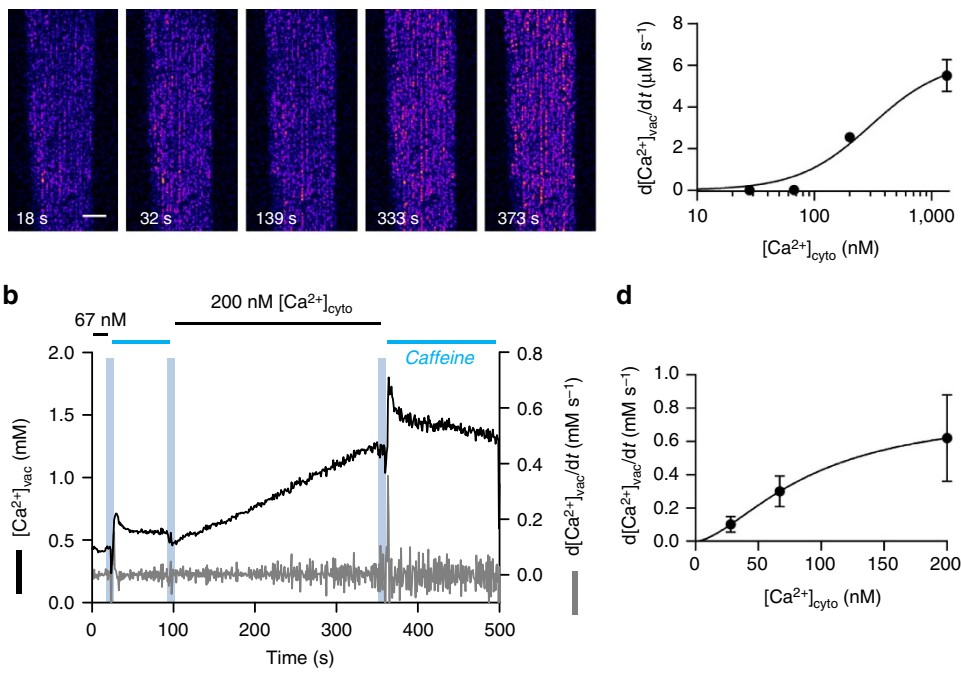

**Figure 5 | Ca$^{2+}$ handling by t-system vacuoles of human skeletal muscle.** (**a**) Ca$^{2+}$ uptake by vacuoles present in the t-system upon exposure to caffeine or 67 nM free [Ca$^{2+}$]$_{cyto}$ and 200 nM [Ca$^{2+}$]$_{cyto}$. (**b**) Representative trace of Ca$^{2+}$ uptake by the vacuoles in a human fibre after exposure to 67 nM free [Ca$^{2+}$]$_{cyto}$, caffeine and 200 nM [Ca$^{2+}$]$_{cyto}$. The vertical blue bars indicate the timing of the internal solution exchanges and the horizontals bars at top indicate the [Ca$^{2+}$]$_{cyto}$ of the standard solution or the application of caffeine to deplete the SR of Ca$^{2+}$. (**c**) Ca$^{2+}$ uptake rate of the vacuoles during exposure to the [Ca$^{2+}$]$_{cyto}$ indicated on the x-axis. (**d**) Ca$^{2+}$ uptake rate of the vacuoles during caffeine-induced Ca$^{2+}$ release following SR Ca$^{2+}$ loading at the [Ca$^{2+}$]$_{cyto}$ indicated on the x-axis. The plots in **c**,**d** were constructed from 15 and 16 values; 6 and 6 fibres; and 3 and 3 subjects, respectively. The data points are represented as mean ± s.e.m. The $r^2$ values for the fits in **c**,**d** are 0.931 and 0.629, respectively. Scale bar: 25 µm.

In some fibres exposed to 1.3 µM [Ca$^{2+}$]$_{cyto}$, vacuoles were observed to form, as in this fibre obtained from a recreationally active individual (Fig. 4). The transverse pattern of the t-system is clear in the first image, marked 110 s, where the fibre was bathed in an internal solution containing 67 nM [Ca$^{2+}$]$_{cyto}$. The addition of caffeine caused the activation of SOCE and the depletion of the [Ca$^{2+}$]$_{t-sys}$ (imaged marked 176 s). However, the application of [Ca$^{2+}$]$_{cyto}$ at 1.3 µM caused the t-system to change its structure over the course of the next ~100 s (Fig. 4). In addition to the increased Ca$^{2+}$-dependent fluorescence emitted from the transverse tubules, the images marked 221, 254, 273, 296 and 316 s show that vacuoles formed in the t-system. The longitudinal orientation of the vacuoles is consistent with these structures forming from longitudinal tubules[16] (Figs 1 and 2).

The response of the partially vacuolated t-system to store-depletion is shown in the image marked 587 s in Fig. 4. The transverse tubules of the fibre were depleted of Ca$^{2+}$ indicating that they were able to conduct SOCE but the vacuoles retained high levels of [Ca$^{2+}$] indicating that: (i) the vacuoles cannot conduct SOCE; and (ii) the luminal connection between vacuoles and transverse tubules must be significantly restricted as vacuole Ca$^{2+}$ is not lost to the cytoplasm via diffusion into the transverse tubules and exit through the chronically open store-dependent channels.

The overall Ca$^{2+}$-dependent fluorescence signal from the t-system progressively increases over the ~200 s in 1.3 µM [Ca$^{2+}$]$_{cyto}$ and the restricted access between the lumen of the transverse tubules and vacuoles indicates that Ca$^{2+}$ was being continuously sequestered from the cytoplasm across the vacuole membrane. We can conclude from these observations that the absolute amount of Ca$^{2+}$ held by the t-system in the presence of vacuoles is significantly increased, probably by a factor close to

the increase in volume of the t-system under vacuolation (Fig. 2b).

The incidence of acute vacuole formation within minutes of exposure to high [Ca$^{2+}$]$_{cyto}$ was 5/15 fibres. The presence or absence of myosin ATPase inhibitors did not affect the formation of vacuoles, indicating that contraction is not involved in t-system vacuolization. The exposure of the t-system to 5 mM [Ca$^{2+}$]$_{cyto}$ and ionomycin caused 18/29 fibres to produce vacuoles. In ionomycin-treated fibres, in every case vacuolation could be rapidly reversed by the removal of Ca$^{2+}$ (Supplementary Fig. 6).

In fibres with a large proportion of vacuoles we tracked [Ca$^{2+}$]$_{t-sys}$ ($t$) during SR Ca$^{2+}$ release and during exposure of the fibres to different [Ca$^{2+}$]$_{cyto}$ to assess the Ca$^{2+}$-handling properties of vacuoles. In largely vacuolated fibres the transverse tubules could not be discriminated in confocal images (Fig. 5a). This may be partly because the vacuoles sequestered the majority of the t-system trapped rhod-5N and partly because the detectors of the confocal microscope were optimized for the stronger fluorescence signal from the vacuoles (the signal from the relatively low [rhod-5N] of the transverse tubules falls below the detection limit of the confocal microscope with these settings[17]). Thus transverse tubular [Ca$^{2+}$] is not resolved under these conditions, so we define the signal resolved as vacuole [Ca$^{2+}$] ([Ca$^{2+}$]$_{vac}$). Figure 5a,b show [Ca$^{2+}$]$_{vac}$ ($t$) during: Ca$^{2+}$ release from SR; and exposure to different [Ca$^{2+}$]$_{cyto}$. The release of Ca$^{2+}$ from the SR induced a spike in [Ca$^{2+}$]$_{vac}$ ($t$). The amplitude of the spike was greater following the period where the fibre was bathed in higher [Ca$^{2+}$]$_{cyto}$ (Fig. 5b). The spike in [Ca$^{2+}$]$_{vac}$ following Ca$^{2+}$ release was followed by some slow leakage of Ca$^{2+}$. The [Ca$^{2+}$]$_{vac}$ spike was in contrast to the t-system with no vacuoles, where the [Ca$^{2+}$]$_{t-sys}$ ($t$) was lowered

**Table 1 | Estimate of the calcium content of the t-system before and after vacuolization.**

| Time from exercise (h) | t-sys$_{Vol}$ (%) | $[Ca^{2+}]_{t-sys}$ (relative to t-system volume; mM) | $[Ca^{2+}]_{t-sys}$ (relative to fibre volume; µM) |
|---|---|---|---|
| 0 | 1.4 | 1.4 | 19.6 |
| 24 | 4.8 | 1.5* | 72.0 |
| 48 | 4.4 | 1.5* | 66.0 |

Values for t-sys$_{Vol}$ and $[Ca^{2+}]_{t-sys}$ from Figs 2 and 3, respectively. Note the t-sys$_{Vol}$ is the transverse and longitudinal tubule volumes (1% fibre volume; calculated in pre-exercised fibres) plus the % fibre volume of the vacuoles (Fig. 2b). B$_{t-sys}$ = 1 (ref. 18) and is assumed not to increase with vacuolation. *The $[Ca^{2+}]_{cyto}$ is expected to increase in the post-exercised muscle, causing $[Ca^{2+}]_{t-sys}$ to increase slightly (Fig. 3).

by the activation of SOCE following chronic SR $Ca^{2+}$ release (Fig. 3a,b). $[Ca^{2+}]_{vac}$ (t) could slowly increase under resting conditions when the $[Ca^{2+}]_{cyto}$ was raised above normal resting levels (Fig. 5c). Note that we could observe a composite of vacuoles taking up $Ca^{2+}$ and transverse tubules depleting of $Ca^{2+}$ when both components of the t-system could be spatially discriminated (Supplementary Fig. 7).

**The calcium content of the dynamic t-system.** The absolute amount of $Ca^{2+}$ held in the t-system (expressed per fibre volume) can be estimated as the B$_{t-sys}$ × $[Ca^{2+}]_{t-sys}$ × t-sys$_{Vol}$ (Table 1), where B refers to $Ca^{2+}$-buffering power. The t-sys$_{Vol}$ of the unvacuolated t-system was 1.0% of fibre volume (Supplementary Fig. 3). The increase in % fibre volume (Fig. 2b) was restricted to vacuoles because the t-system trapped dye was drawn into the vacuoles from the transverse tubules (see above). The acute increase in t-sys$_{Vol}$ following training is expected to be largely due to vacuole formation, as the transverse tubules are resistant to volume changes[17,33], so the t-sys$_{Vol}$ post-exercise was calculated as the transverse tubular volume plus the vacuole volume. We conservatively assume that B$_{vac}$ is the same as B$_{t-sys}$[18]. Under this assumption the total calcium held by the t-system increases > fivefold following vacuolation (Table 1). Note that this is conservative estimate of the calcium total held by the t-system, which could be increased by: (i) the value of B$_{vac}$ increasing if diffusible $Ca^{2+}$-buffers enter and persist in vacuoles; and/or (ii) the vacuoles depolarized, causing a more favourable electrochemical gradient for the vacuolar accumulation of $Ca^{2+}$.

## Discussion

In this study we define fundamental properties of the human muscle t-system and how these properties change in response to an acute bout of heavy-load strength training. The t-system was able to vacuolate following either a bout of strength training or application of high resting $[Ca^{2+}]_{cyto}$. The absence of change in the % fibre volume occupied by the vacuoles between 24 and 48 h post-exercise (Fig. 2) suggests the formation of vacuoles is an acute process, occurring soon after the bout of demanding exercise. Vacuoles arose from the longitudinal tubules of the t-system network (Fig. 1; ref. 16) and altered the overall $Ca^{2+}$-handling properties of the t-system. The $Ca^{2+}$-handling properties of the transverse tubules and vacuoles in human muscle were discriminated, showing that: (i) the transverse tubules can respond to store-depletion and increases in $[Ca^{2+}]_{cyto}$ (Fig. 3); (ii) vacuoles only respond to increases in $[Ca^{2+}]_{cyto}$ (Figs 3–5); and (iii) increases in t-system volume via vacuolation increase the absolute amount of $Ca^{2+}$ that the t-system can hold (Table 1). Thus vacuolation (Fig. 2) changes the balance between SOCE and $Ca^{2+}$ uptake across the t-system (Figs 3–5). In addition to this the peripheral arrangement of

vacuoles in the fibre will restrict the diffusion of $Ca^{2+}$ between the interstitial space and the deep t-system (Fig. 1 and Supplementary Figs 2–4). Therefore, we can expect that the amount of $Ca^{2+}$ held inside the fibre will be reduced as it is taken up by the vacuoles. The vacuolated t-system can act as a buffer of $[Ca^{2+}]_{cyto}$, sequestering $Ca^{2+}$ during periods of $Ca^{2+}$ release and at high $[Ca^{2+}]_{cyto}$. We propose the change in muscle plasmalemma structure to buffer $Ca^{2+}$ is important in the prevention or reduction of $Ca^{2+}$-induced damage[34] in the muscle following heavy-load strength exercise, and that this is similar in type I and type II muscle fibres.

We have previously shown that our method of detecting and reconstructing the correct tubule structure and orientation has an accuracy of $\sim 90\%$ (ref. 17). This provides us with a high level of confidence in describing the t-system of human muscle fibres. The fibres obtained from needle biopsies are cut at both ends, excluding the possibility of examining intact fibres. Therefore the well-established approach of imaging the t-system using skinned fibres was employed[17]. We note that mechanically skinning does not affect the structures of the fibre. For example, direct assessment of the sub-sarcolemmal t-system using confocal imaging, super-resolution dSTORM and tomographic electron microscopy presented images of this intricate structure from mechanically skinned fibres that were not distinguishable from how it presented in intact fibre preparations[35].

The 3D reconstruction of the t-system through an entire transverse axis of a human muscle fibre showed the prominent structure of the strength training accustomed t-system was the transverse tubules (Fig. 1). Transverse tubules were dominant in the deeper regions of the fibre. The longitudinally oriented tubules existed entirely at the peripheral regions of the fibre, where sarcomere misalignment occurred (Fig. 1; (ref. 16,36,35)). Our shallow xz projection of the t-system (Fig. 1c) is qualitatively similar to classic manual electron micrograph reconstructions[36]. Furthermore, together with the fact that we only observed high density t-system vacuolation across the 10 subjects biopsied in this study when subjects were sampled at 24 or 48 h post-eccentric exercise or the fibres were exposed to high $[Ca^{2+}]_{cyto}$ (Figs 1–5), we can conclude that t-system in situ structure was not significantly altered by biopsy and mechanical skinning of fibres. We would also fully expect that other structures, such as cytoskeletal component desmin and mitochondria that have been reported to change their orientation in the fibre post-eccentric contractions[37–39] to remain in these positions following biopsy, isolation of fibres and mechanically skinning.

Our observation of longitudinal tubules appearing at the periphery of the fibres, with the sarcomere misalignments is a novel finding that has been possible by our full reconstruction of the human muscle fibre t-system (Fig. 1). The t-system must navigate around sarcomere misalignments and around local structures, such as nuclei[16,17] to maintain communication of the plasmalemma through all depths of the fibre. Sarcomere misalignments have been a regular observation across vertebrate skeletal and cardiac muscle[16,17,33,36,40], suggesting that a degree of sarcomere misalignment is the basal condition in this tissue. The density of misalignments can increase following damage to myofibril arrangements, especially in conditions such as muscular dystrophy[41,42]. The misalignment of sarcomeres at the fibre periphery of healthy muscle may be a normal part of new myofibrils being laid down during normal turn-over. Misalignment would occur when the resting sarcomere length of the new myofibrils is different from that of older myofibrils. The accumulation of small heat shock protein in disrupted areas, in fibres not totally damaged by eccentric exercise, also points at the structures in the periphery of the fibres as the most vulnerable to exercise-induced disruptions[43].

The formation of vacuoles post-exercise provided large pockets of the t-system where $Ca^{2+}$ could be sequestered or accumulated. The diameter of the vacuoles is approximately an order of magnitude greater than the mean tubule width estimated by fluorometric calibration method (Supplementary Fig. 3), which we established for vertebrate skeletal muscle in a previous study[17]. This figure therefore translates to a ~100-fold increase in the local tubule volume during vacuolation. The increase in t-sys$_{Vol}$ associated with vacuolation (Fig. 2) caused accumulation of the t-system trapped small fluorescent molecule (fluo-5N) in these compartments, as previously observed in toad[16]. The shift in the finite amount of dye in the sealed t-system from the transverse tubules caused patchiness of signal within these structures, which can be attributed to the fluorescence signal falling below the detection limit[17]. The low transverse tubular fluorescence signal also excluded the possibility of reconstructing the vacuolated t-system in 3D. The strong fluo-5N fluorescence signal emitted from the vacuoles and low signal from the transverse tubules was consistent with accumulation of $Ca^{2+}$ in the vacuoles at the expense of $Ca^{2+}$ previously contained in the transverse tubules (Fig. 2). It is also noteworthy that there is a heterogeneous change in sarcomere alignment after eccentric exercise[44]. However, it is difficult to quantify whether there are more sarcomeres misaligned along with vacuolation because our reference, the transverse tubules, become much less transversal in the post-exercised muscle (Fig. 2e).

In addition to the apparent passive redistribution of $Ca^{2+}$ and small molecules in the t-system, active processes causing $Ca^{2+}$ to be sequestered in vacuoles was observed (Fig. 5). Vacuoles took up $Ca^{2+}$ rapidly during periods of SR $Ca^{2+}$ release and more slowly when $[Ca^{2+}]_{cyto}$ was above normal resting levels in the absence of $Ca^{2+}$ release (Fig. 5), even when diffusion between the transverse tubules and vacuoles appeared highly restricted (Fig. 4). We note the presence of vacuoles themselves require the presence of functional $Na^{+}$-$K^{+}$ ATPases[13,25]. Thus the maintenance of high levels of $Ca^{2+}$ in the t-system in the days post-eccentric exercise is an energy requiring process. Consistent with this, a depletion of muscle glycogen has been observed in human muscle late in recovery (~48 h) from eccentric exercise[39]. The $Ca^{2+}$-handling proteins of the vacuole membrane are likely to be the NCX and PMCA that have previously been described to translocate $Ca^{2+}$ into the skeletal muscle t-system[45–47].

The absence of SOCE activity and increased $Ca^{2+}$-uptake ability of vacuoles causes a net shift of $Ca^{2+}$ from the intracellular environment of the fibres to the vacuolar lumen. Our estimation of an increase in total calcium content of the t-system upon vacuolation may be a low one because it does not include the possibility that diffusible $Ca^{2+}$-buffers may occupy the vacuoles to increase the $Ca^{2+}$-buffering power of the t-system (Table 1) or the possibility that vacuoles may be depolarized, creating a favourable electrochemical gradient for $Ca^{2+}$ accumulation. The vacuolated t-system sequesters (conservatively) 8–9% of the total fibre calcium (Table 1; ref. 31), making this $Ca^{2+}$ unavailable to the SR and cytoplasm. The 'trapping' of the $Ca^{2+}$ within the vacuoles in the presence of chronically activated SOCE may be assisted by tight restrictions, significantly slowing diffusion at the luminal junction between the transverse tubules and vacuoles[16].

The changing structure of the vacuoles over days (Fig. 2) and the rapid devacuolation of the fibre observed in ionophore and 0 $Ca^{2+}$ (Supplementary Fig. 6) indicates the t-system likely responds to changing levels of $[Ca^{2+}]$. The increasing roundness of vacuoles 48 h post-exercise (Fig. 2) could reflect a mechanism of membrane repair. For example, dysferlin-mediated constrictions in the membrane could fragment and turn over vacuolated longitudinal tubules in a $Ca^{2+}$-dependent manner[48].

The clear segregation of functional SOCE at the transverse tubules versus the vacuoles is consistent with SOCE being conducted rapidly across the transverse tubule by Orai1 in conjunction with SR terminal cisternae STIM1L[18–20,22,32,49,50], and the fact that skeletal muscle lack the molecular machinery to form junctions with longitudinal tubules spanning the sarcomere[51]. Because muscle fibres can never be depleted of $Ca^{2+}$ under physiological conditions[21,52], the physiological activation of SOCE must occur in fibres that are full of $Ca^{2+}$ (ref. 52). The activity of the RyRs likely creates $Ca^{2+}$ gradients that are essential to the activation of SOCE under physiological conditions[18]. This scenario limits SOCE activation to junctional membrane regions that possess RyRs, thus excluding vacuoles from conducting SOCE regardless of Orai1 and STIM1 localization[49].

The change in $Ca^{2+}$-handling properties as the t-system vacuolates is due to a shift in the balance of $Ca^{2+}$ movements across the t-system, where the presence of vacuoles skews the net movement of $Ca^{2+}$ towards extrusion from the fibre. This outcome was dependent on the absence of SOCE activity in the vacuoles and the accumulation of $Ca^{2+}$ within the vacuoles from both the fibre cytoplasm and from the transverse tubular lumen. We provide the first evidence that the t-system may be able to regulate the flow of ions from the interstitial space by changing its structure close to the fibre periphery (Figs 1 and 2; Supplementary Figs 1–4). The vacuolization process and accumulation of $Ca^{2+}$ paralleled the onset and decline of muscle soreness, which is commonly associated with eccentric exercise[27]. The appearance of vacuoles increased the $Ca^{2+}$-content of the t-system (Fig. 2) at the expense of $Ca^{2+}$ available to the fibre for contraction (Figs 2–5). We suggest that this is an adaptive response to avoid damage otherwise initiated by increases in $[Ca^{2+}]_{cyto}$ (ref. 34) or the mechanical stress of tension development. The outcome for the muscle would be a majority of fibres remaining viable following the cessation of DOMS, as observed, for example, in long-distance runners[11,12]. To the best of our knowledge we provide the first evidence of a plasmalemma significantly changing its structure to change its $Ca^{2+}$-handing properties to adapt to a physiological stress. Finally, we have also discriminated the $Ca^{2+}$-handling properties of the t-system in human skeletal muscle fibres for the first time (Figs 3–5). While observing the expected fibre heterogeneity of the human *v. lateralis*, no significant differences in t-system $Ca^{2+}$-uptake rates were observed between the fibre types. This is in contrast to the clear delineation of these t-system properties between the glycolytic fibre types (IIb/x fibres) from the oxidative fibre types (type I and IIa) of rodents[18,53].

## Methods

**Exercise bouts.** All participants who undertook exercise bouts were men between the ages of 22 and 26 years with over 12 months of strength training experience. Exercise bouts were *traditional* for strength training with respect to containing multiple sets, combining concentric and eccentric contractions of the lower body (see Roberts *et al.*[26] for a detailed breakdown). The exercise significantly increased surrogate markers of cell membrane disruption, with systemic increases in myoglobin ≤ 24 h post-exercise ($P < 0.05$; fivefold peak increase) and creatine kinase activity ≤ 48 h post-exercise ($P < 0.05$; 1.4-fold peak increase), suggesting muscle damage occurred[26]. The fibres collected from this group were used to examine the structural detail of the t-system before and after exercise. The exercise protocols are provided in detail in the Supplementary Methods. A second group of subjects used for resting muscle biopsies were collected from recreationally active men ($n = 5$, 18–42 years old) and women ($n = 2$, 38 and 45 years old). The fibres collected from this group were used to assess the $Ca^{2+}$-handling properties of the t-system. All procedures were approved by the University of Queensland Human Ethics Committee and informed consent was received from all participants prior to commencement of their involvement in this study.

**Muscle biopsies and preparation for single fibre imaging.** Muscle biopsies were collected under local anaesthesia (Xylocaine, 10 mg ml$^{-1}$ with adrenalin, 5 μg ml$^{-1}$) from the mid-portion of the *vastus lateralis* muscle, using a 6-mm

Bergstrom biopsy needle modified for manual suction[26]. Biopsies were collected from three separate incisions, each ~3 cm proximal to the previous to avoid multiple-biopsy influences on the collected tissue.

Muscle tissue collected from the biopsy needle was blotted on filter paper (Whatman No 1) to remove blood and external fluid. The muscle tissue was then placed in a Petri dish under a layer of paraffin oil[54,55]. A bundle of fibres were isolated and exposed to a physiological solution containing (mM): NaCl, 145; KCl, 3; CaCl$_2$, 2.5; MgCl$_2$, 2; fluo-5N salt, 10; or rhod-5N salt, 2.5; BTS (Calbiochem), 0.05; and HEPES, 10 (pH adjusted to 7.4 with NaOH). Fibres were allowed >10 min to equilibrate with the physiological solution and then individual fibres were isolated and mechanically skinned. Skinned fibres with t-system-trapped fluorescent dye were mounted on a custom-made chamber that used a coverslip as a base and bathed in a standard internal solution, which contained (mM): EGTA, 50; Hepes, 10; K$^+$, 126; Na$^+$, 36; ATP; 8; Mg$^{2+}$, 1; creatine phosphate, 10; Ca$^{2+}$, 6.7 × 10$^{-4}$. [Ca$^{2+}$] in the standard internal solution was varied in the range 28 nM to 1.3 μM. Ca$^{2+}$ was released from the SR using a similar solution in the nominal absence of Ca$^{2+}$, 0.01 mM Mg$^{2+}$ and 30 mM caffeine. Fibres with t-system trapped rhod-5N, rhod-5N fluorescence and [Ca$^{2+}$]$_{t-sys}$ were calibrated by permeabilizing the t-system to Ca$^{2+}$ with ionophore and consecutively introducing solutions containing 5 mM and 0 Ca$^{2+}$ (ref. 18). Fibres that were t-system trapped fluo-5N were used for high spatial resolution imaging of the t-system (see Jayasinghe et al.[35]). Fibres used for imaging were assessed for myosin heavy chain isoform[54,55] using western blotting methods described previously.

**Confocal imaging.** Mounted skinned fibres were imaged using an Olympus FV1000 confocal microscope equipped with an Olympus 1.3NA 63x Plan-Apochromat objective. Fluo-5N and rhod-5N dyes were excited with a 488 nm Ar-ion laser and 543 nm HeNe laser, respectively, and the emission was filtered using the Olympus spectra detector. In 3D imaging for reconstructing the fibre structure, the pinhole was adjusted to 0.7 airy units and each confocal section was averaged over 2–4 sweeps. Both the zoom and z-stepping were set to achieve a voxel size finer than 90 nm in x and y and 150 nm in z. All images were recorded onto 640 × 640 pixel 16-bit TIFF format images. For tracking Ca$^{2+}$ movements across the t-system membrane images were continuous recorded in xyt mode with an aspect ratio of 256 × 512, with the long aspect of the image parallel with that of the preparation. Temporal resolution of imaging in this mode where the fluorescence signal emanated from within the borders of the fibre was 0.8 s.

*Image analysis for 3D imaging.* Confocal z-stacks were subject to a Richardson-Lucy maximum-likelihood iterative deconvolution with the primary objective of reducing noise in the image but with an additional benefit of improved contrast and the effective resolution as detailed previously[17,35]. The point spread function of the imaging system was estimated by averaging the images acquired under the same settings of 15 polystyrene microspheres (100 nm ⌀) immersed within the same internal solution for imaging the fibres. Both the deconvolution and point spread function analysis were implemented in IDL programming language (ExelisVis). Deconvolved image volumes were subjected to a 3D skeletonizing algorithm implemented in Amira 5.4 (Visage Imaging, Germany) and the skeletons were surface rendered for further visualization through MayaVi2 data visualizer implemented in Python. Single planes sectioning through the middle of the fibre in deconvolved confocal volumes were analysed with the 'Directionality' plug-in in FIJI (ImageJ). This analysis produced percentage histograms of the local tubule orientation in relation to the fibres' transverse plane, as demonstrated previously[35]. For quantifying the spatial properties of vacuoles, raw confocal volumes were opened in FIJI (ImageJ) programme. The high fluorescence intensity of vacuoles was exploited for determining an intensity threshold (typically at the 50–60th percentile of the intensity histogram) capturing only the vacuolated regions. The resulting binary mask was analysed with the 'Particle Analyzer' plug-in to (i) directly count the number of vacuoles within the confocal volume, (ii) calculate the fraction of the fibre volume that is occupied by vacuoles and (iii) the 'roundness' of each vacuole, calculated in each plane by fitting to each vacuole region a 2D ellipse whose width was divided by the length (that is, to calculated the inverse of the aspect ratio).

*Image analysis for Ca measurements.* t-system rhod-5N fluorescence (t) (F (t)) was collected during continuous xyt imaging during multiple internal solution changes. At the end of the experiment each fibre was exposed to ionophore and 5 mM Ca$^{2+}$, followed by 0 Ca$^{2+}$ to obtain the fluorescence maximum (Fmax) and minimum (Fmin), respectively. These values were used in conjunction with the previously determined $K_D$ of rhod-5N in the t-system of 0.8 mM (ref. 18) to determine [Ca$^{2+}$]$_{t-sys}$, with the relationship: [Ca$^{2+}$]$_{t-sys}$ (t) = $k_{D,Ca}$ × (F(t) − Fmin)/(Fmax − F(t)).

Note that image analysis for t-system structure and Ca$^{2+}$-handling was performed blinded, that is without knowledge of type of exercise or fibre type.

Data are presented as mean ± s.e.m. Statistical analysis was performed with Graph Pad Prism. All data are available from the authors, on request.

**Data availability.** The data that support the findings of this study are available from the corresponding author upon reasonable request.

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

## Acknowledgements

We thank the individuals who volunteered as subjects in this study. This work was supported by an Australian Research Council (ARC) Discovery Project (DP110102849) and a National Health & Medical Research Council Project Grant (GNT1025355) to B.S.L.; and Royal Society (UK) grant (RG.IMSB.107729) and Wellcome Trust grant (RG.IMSB.104532.054) to I.J. B.S.L. was a Future Fellow of the ARC (FT140101309).

## Author contributions

T.R.C. designed, performed and analysed experiments; R.M.M. performed and analysed experiments; L.R., designed and performed experiments; T.R. designed and performed experiments and performed biopsy; R.G.F. performed biopsy; J.S.C. designed experiments; I.J. designed, performed and analysed experiments, and wrote the paper; and B.S.L. designed, performed and analysed experiments, and wrote the paper.

## Additional information

**Competing financial interests:** The authors declare no competing financial interests.

