## [Peer Review File · Nature Communications]

Reviewers' comments:

Reviewer #1, expert in store operated calcium entry (Remarks to the Author):

There can be no doubt that skeletal muscle responds to different types of exercise in order to remodel the responsiveness of subsequent bouts of exercise. This involves short and long term changes to the muscle properties involving fibertype, mitochondrial content and Ca²⁺ handling. Because of the importance of exercise to human health and well-being, a full understanding of mechanism underlying exercise adaptation is likely to offer important insight as well as novel therapeutic targets. In the current work, Cully et al introduce a novel component to the exercise response in muscle. They test the idea that T-system remodels in response to intense exercise in order to shape Ca²⁺ handling. Using sophisticated 3D-confocal imaging and Ca²⁺ imaging, the authors show that exercise induce in a modulation of the longitudinal tubules into vacuoles in a Ca²⁺ dependent manner. The vacuoles can then alter the Ca²⁺ handling properties of the muscle fiber. While this is really creative work done in humans, it is preliminary and would require additional work.

-The exercise protocol involved an intense eccentric exercise that led to membrane damage and serum myoglobin elevation. How does the disruption of the membrane effect the skinned fiber data? The numbers of subjects seem very low. Was this based on a power analysis? How percentage of fibers were utilized? What were the criteria to exclude a fiber? Do other forms of exercise also create vacuoles? Is there a spectrum or is this process limited to extreme types of exercise.

-Much of the techniques used involved the skinned fiber technique. Certainly this technique provide key physiologic Ca²⁺ measurements and is well validated, but to form hypothesis based on structural information it is important to validate findings using intact muscle fibers. It would be ideal to express a genetically encoded Ca²⁺ indicator that localizes to the TT. One could then assess maneuvers to mobilize this pool in intact muscle fibers. This might really provide compelling information for the vacuole formation.

-Exercise induces many changes to the fiber including cytoskeleton, mitochondria and other organelles like golgi. It is important to put these structural changes into that context especially since it is unclear how the skinning of fibers alters the relationship of TT to mitos, cytoskeletal etc.

-The authors mention that differences in fiber types might influence the results. It seems that this is an important issue given that Ca²⁺ handling differs between fiber types. It seems knowing whether vacuoles are formed in slow v fast or vice versa is important to this phenomenon.

-What is the relationship of the mitochondria and vacuoles? Is the Ca handling energy requiring?

-Since the vacuoles are rather large, can they be isolated (differential centrifugation) using standard biochemical techniques. This would enable a more thorough characterization of the vacuoles offering important validation of the imaging studies. Are there differences in the expression of proteins b/n the vacuoles and TT?

In skeletal muscle the tight control of cytosolic calcium is critical to maintain proper cellular function. When cytosolic calcium is in excess, it drives oxidative stress and calpain proteases, which disrupt signaling pathways and EC coupling structures.

While most in the field have focused the mechanisms for calcium associated muscle dysfunction in disease, we still have critical gaps in our understanding of how calcium is managed in the healthy skeletal muscle cell; especially during and after severe fatiguing exercise or injurious muscle contractions. Revealing these mechanisms in healthy muscle is critical to *fully understand* why calcium handling is dysregulated in disease.

This manuscript by Culley et. al. offers timely and intriguing evidence that the functional remodeling of the t-tubular system is a central player in how the skeletal muscle manages cytosolic $[Ca^{2+}]$ after a bout of un-accustomed exercise. Specifically, these authors demonstrate that the longitudinal t-tubule system is a privileged participant in this process by forming vacuoles that sequester Ca^{2+} . Support is provided using a host of advanced imaging techniques to quantify T-tubular morphology and function. The use of biopsy material from human participants pre- and post-exercise is a powerful approach to address this question.

While this reviewer finds significant merit in this work, there are several areas in this manuscript that may diminish its impact and accessibility to the field.

Major Comments:

1. Taken together the results shown support a model in which the vacuolization of the longitudinal t-system (LT) is a 'saftey net' by which the muscle cell can sequester excess myoplasmic Ca^{2+} associated with exhaustive exercise. If successful, the muscle cell would be able to maintain it viability while it recovers from this insult.
 - a. "If" this is an accurate summary of the major results, the introduction should be re-tooled to better set-up this concept. For example, the statement; Line 93: "*A compartmentalization of Ca^{2+} in vacuoles would be an advantage in the days following heavy-load resistance exercise, when delayed onset muscle soreness and low-frequency or long-duration fatigue are experienced (24).* These vacuoles would have protected the muscle such that 'fatigue' was not irrecoverable. Rather it is reversible with muscle soreness and long duration fatigue being a consequence.
 - b. Line 86: "*Vacuole formation has been associated with fatigue and lactate movements in mouse muscle. (17, 18) but the physiological importance of vacuolar accumulation of small molecules or ions following exercise has not been explored.*" If vacuole formation has been studied in fatigue, it has been studied during 'exercise'. Please rephrase.

- a. Line 96: The sentence “*To provide evidence for a role of the t-system as a dynamic Ca²⁺ buffer.....*” Suggests that this experiment was not hypothesis driven, rather an experiment to confirm an established phenomenon. As I trust this is not the author’s intention, please revise.
2. Line 177: The authors state: “*regions abundant of longitudinal tubules were identified; often these extended as a series of longitudinal tubule networks across multiple sarcomeres (Fig 1B), typically in between myofibrils that lack alignment between sarcomeres (suppl Fig S1 E&F)*”
 - a. Can the authors elaborate on the finding that there are ‘regions’ of these LT’s?. If these are seen in heterogeneous localization, this is likely not a physiologically relevant structure at baseline and rather a consequence of an adaptation.
 - b. The idea that these LT’s occur at areas of sarcomere/myofibrillar miss-alignment may suggest that they have arisen secondary to a damage and repair consequence (see ^{1,2}). This can occur due to physical damage or activation of calpains that cleave desmin which hold these myofibrillar structures together. In fact, there is an intriguing hypothesis that this desmin cleavage may also be a ‘safety net’ that prevents damage with exhaustive exercise (see ³⁻⁵).
 - c. The authors state that the biopsies were collected in the basal condition ‘prior to exercise’.
 - i. Which group was used to support the statement in Line 177? The recreational athlete? The trained exercisers? How long prior to the last bout of exercise was the biopsy taken?
 - ii. It may be important to comment on the basal morphology of both groups (novice and trained) as differences in their occurrence may provide insight.
 - iii. Did the authors note more sarcomere miss-alignments along with the vacuolization in the muscle fibers after exhaustive exercise?
3. The authors are obliged to provide more detail regarding the origination of the muscle samples.
 - a. These authors need to state implicitly that this current study used samples from a recently published study. They then need to provide details on the subjects and their exercise regimes. If its duplicative, put it in supplemental methods as is not appropriate to reference a published 2015 paper in which muscle biopsies were taken from multiple groups and treatment conditions.

- b. In fact this is potentially perilous to piggy back on this 2015 study as they focused on a 'rescue' condition (cold treatment) that should reduce the amount of vacuoles if these authors hypothesis is true. Do these authors have this data?
- 4. Line 285: The authors state "*The 'trapping' of the Ca²⁺ within the vacuoles in the presence of chronically activated SOCE, may be assisted by funnel-like collars at the luminal junction between the transverse tubules and vacuoles (9). Such structures would maintain a pathway for the entry of Ca²⁺ into the vacuoles from the transverse tubules, but make the exit of Ca²⁺ through a very small aperture an extremely slow process.*"
 - a. The authors appear to be taking liberties with the structures they are defining. In the 2008 paper (ref 9) the authors were describing the movement of ions in the LT and suggesting they moves slower based on restricted space. Here the authors suggest that vacuoles are cut-off from the rest of the t-system. If that is the case, there should be no tunneling into a vacuole, rather there should be active transport into this vacuole.
 - i. Is this the case? If so, can the authors comment on the means by which the calcium is transported in?
 - ii. How does the calcium accululate to such a great extent in the vacuole? Is their trapped buffer inside the vacuole to allow calcium to accumulate to the great extent that is predicted?
 - b. Can the authors speculate as to why this vacuolization occurs primarily near the muscle fiber surface? And not throughout the myofiber?

1. Head SI, Stephenson DG, Williams DA. Properties of enzymatically isolated skeletal fibres from mice with muscular dystrophy. *J Physiol*. 1990;422:351–367. Available at: http://www.ncbi.nlm.nih.gov/entrez/query.fcgi?cmd=Retrieve&db=PubMed&dopt=Citation&list_uids=2352184.
2. Buttgereit A, Weber C, Garbe CS, Friedrich O. From chaos to split-ups - SHG microscopy reveals a specific remodelling mechanism in ageing dystrophic muscle. *J Pathol*. 2013;229:477–485. doi:10.1002/path.4136.
3. Sam M, Shah S, Fridén J, Milner DJ, Capetanaki Y, Lieber RL. Desmin knockout muscles generate lower stress and are less vulnerable to injury compared with wild-type muscles. *Am J Physiol Cell Physiol*. 2000;279(9151):C1116–C1122.

4. Shah SB, Davis J, Weisleder N, et al. Structural and functional roles of desmin in mouse skeletal muscle during passive deformation. *Biophys J*. 2004;86(5):2993–3008. doi:10.1016/S0006-3495(04)74349-0.
5. Lieber RL, Shah S, Fridén J. Cytoskeletal disruption after eccentric contraction-induced muscle injury. *Clin Orthop Relat Res*. 2002;(403 Suppl):S90–9. Available at: <http://www.ncbi.nlm.nih.gov/pubmed/12394457>. Accessed March 9, 2014.

Reviewers' comments:

Reviewer #1, expert in store operated calcium entry (Remarks to the Author):

There can be no doubt that skeletal muscle responds to different types of exercise in order to remodel the responsiveness of subsequent bouts of exercise. This involves short and long term changes to the muscle properties involving fibertype, mitochondrial content and Ca²⁺ handling. Because of the importance of exercise to human health and well-being, a full understanding of mechanism underlying exercise adaptation is likely to offer important insight as well as novel therapeutic targets. In the current work, Cully et al introduce a novel component to the exercise response in muscle. They test the idea that T-system remodels in response to intense exercise in order to shape Ca²⁺ handling. Using sophisticated 3D-confocal imaging and Ca²⁺ imaging, the authors show that exercise induce in a modulation of the longitudinal tubules into vacuoles in a Ca²⁺ dependent manner. The vacuoles can then alter the Ca²⁺ handling properties of the muscle fiber. While this is really creative work done in humans, it is preliminary and would require additional work.

We thank the reviewer for recognising the novelty of our work. He/she has raised a number of points which we have addressed in revising the manuscript. We have also demonstrated that we have indeed sampled the ideal number of biopsy fibres to demonstrate the effects of eccentric exercise for the observations to be robust. The reviewer suggests approaches such as genetically-encoded Ca sensors and differential fractionation of vacuole membranes or compartments which are proven approaches in studying other cell types. As mentioned below we point out that these methods are not feasible for analysing human muscle biopsies specifically for studying exercise-induced vacuoles due to the inherent properties of skeletal muscle fibres. We also point out intact fibre experiments in the revised manuscript in published studies where mice have been used.

-The exercise protocol involved an intense eccentric exercise that led to membrane damage and serum myoglobin elevation. How does the disruption of the membrane effect the skinned fiber data?

An important advantage of using skinned muscle fibres is that fibres with any significant membrane damage would rapidly loose the dye from the sealed t-system in a matter of seconds due to the very large surface to volume ratio of the sealed t-system. Therefore, fibres with membrane damage would self-discard from experimentation. Note that the imaging protocols used in this study lasted for up to 45 minutes with no observable drop in dye content that was not accountable by photo-bleaching.

The numbers of subjects seem very low.

There were 7 subjects for the Ca²⁺ handling experiments (viewing t-system structure and tracking t-system Ca movements in recreationally active people, no formal training); 6 for the chronic training (viewing t-system structure 4-5 days before and 6-7 days following the last training session of the 12 week training program) and 3 for the acute training (viewing t-system structure before, 24 & 48 hrs

after unaccustomed, eccentric exercise). The combination of all three sets of experiments shows that from 16 subjects (>80 fibres), vacuole density was only high when t-system structure was imaged 24 & 48 hrs post unaccustomed eccentric exercise, or on occasions when the skinned fibres were exposed to high $[Ca^{2+}]_{cyto}$. The results are statistically significant (see Fig 2 legend; also see Discussion, p.9, para 3). Also note that the University of Queensland Human Ethics Committee imposed a requirement that we use the minimum number of subjects to achieve statistical significance in the study since there is risk of infection with each biopsy taken and our participants had up to 8 biopsies each.

Was this based on a power analysis?

A power analysis was not necessary.

How percentage of fibers were utilized?

We apologize, but we are not entirely sure whether the reviewer is asking what percentage of fibres are used from a collected biopsy or whether he/she is asking what percentage of fibres that we imaged were included in the analysis. All fibres that were imaged have been included in the data presented in the manuscript. It is accepted that a muscle biopsy from a small region of the large vastus lateralis muscle is representative of that muscle for biochemical determinations (although not for determining fibre type distribution, Nygaard and Sanchez, Anat Rec, 1982). Our advantage is that we actually sample more than one fibre from the biopsy, so have repeat measures from the same biopsy from a given time point. This is now stated at the start of the Results section.

What were the criteria to exclude a fiber?

All fibres imaged were included in the dataset. As indicated above, a major advantage in using skinned muscle fibres is that fibres with any significant membrane damage rapidly lose the dye from the sealed t-system in a matter of seconds and therefore, self-exclude from further experimentation.

Do other forms of exercise also create vacuoles? Is there a spectrum or is this process limited to extreme types of exercise. It seems that only eccentric exercise causes vacuoles, this can include resistance exercise (as in our study) or long distance running. Following long distance running there is an accumulation of Ca in the muscle and damage to some fibres in the muscle. These type of exercises are now referenced in the revision in regards to the accumulation of Ca and eccentric contraction induced damage and the important observation that only a small % of examined fibres in human studies are damaged (thus there appears to be some protection against Ca-induced damage). Additionally, study in mouse showed that concentric contraction, the other major type of exercise, failed to produce vacuoles while eccentric contractions (10 tetani simultaneous with stretch) were sufficient to generate vacuoles (Yeung et al 2002, J.Physiol). This information is now presented in the revision (see Introduction, p.3, para 4.).

-Much of the techniques used involved the skinned fiber technique. Certainly this technique provide key physiologic Ca²⁺ measurements and is well validated, but to form hypothesis based on structural information it is important to validate findings using intact muscle fibers. It would be ideal to express a genetically encoded Ca²⁺ indicator that localizes to the TT. One could then assess

maneuvers to mobilize this pool in intact muscle fibers. This might really provide compelling information for the vacuole formation.

As the reviewer indicates, the skinned fibre technique has indeed been validated extensively – in our experience that is for both function and structure. The architectures of both longitudinal and transverse tubules observed in early EM studies (e.g. Peachy & Eisenberg 1978, Franzini-Armstrong et al 1999) were faithfully reproduced in the 3D reconstructions of the t-systems using the skinned-fibre method (Jayasinghe & Launikonis 2013). We have also performed a direct assessment of the integrity of t-system elements using other techniques such as tomographic electron microscopy and super-resolution dSTORM. Indeed, even the fine sub-sarcolemmal tubular network remains completely intact following mechanically skinning, indistinguishable from how it presents in intact fibre preparations (see Jayasinghe et al 2013, *Biophys J*). This point is now raised in the revised manuscript. Furthermore, as mentioned above, Yeung et al have previously shown eccentric contractions in mouse muscle induces vacuoles, whereas based on imaging intact fibres, concentric contractions do not. This study is now discussed in the revised ms. We must underscore here, though, that our study is the first to show: (i) vacuolization in human muscle fibres within specific regions of the fibres consisting of abundant longitudinal tubules; (ii) vacuoles persist over days in parallel to the transient of DOMS; (iii) a differentiation of Ca^{2+} handling properties of the vacuoles and transverse tubules; (iv) is the first to describe the t-system Ca-handling properties in human; and, finally (v) is the first to present the role of vacuoles in muscle fibres is to prevent Ca-induced damage during the post-exercise period of time that the muscle is highly vulnerable.

Expressing a targeted sensor (Ca-sensitive or otherwise) in the t-system or connected structures would in principle provide useful insights into the mechanisms underpinning the vacuolation; it could also facilitate elegant experimental approaches to studying the working muscle. However, we are bound by the ethical and technical constraints in expressing genetically encoded Ca indicators in human muscle. This could only be performed in fibres cultured post-biopsy, then exposed to virus to express a Ca indicator. Firstly, human muscle biopsy techniques which reliably produce the quality of intact differentiated muscle fibres that is required for primary culture do not yet exist. Even if such a protocol were to be developed, there is no evidence that the t-system structure is preserved *in vitro* over the timescales required for expressing Ca-sensor transgenes. In fact, the data presented of the follow-up biopsies underscore the intrinsic properties of the t-system to undergo remodelling and self-repair in the **days** that follow exercise, a feature that is not compatible with any type of post-hoc genetic manipulation of the fibre. Therefore, such an approach is unsuitable for validating the structural and functional integrity of the t-system in skinned fibres.

-Exercise induces many changes to the fiber including cytoskeleton, mitochondria and other organelles like golgi. It is important to put these structural changes into that context especially since it is unclear how the skinning of fibers alters the relationship of TT to mitos, cytoskeletal etc.

Again, there is no evidence that skinning fibres changes the structural relationship between internal structures within the fibre. All the evidence is to the contrary, as mentioned above (eg. Jayasinghe et al 2013, *Biophys J*; Jayasinghe & Launikonis, 2013, *J Cell Sci*). Reviewer 2 actually makes the point that desmin (cytoskeletal protein) is lost rapidly from muscle with eccentric exercise, leading to poor sarcomeric organisation. We now raise in the Discussion (p. 9, para 3) that other structures that are

known to change their positioning with eccentric contractions to be expected to be in the same position in skinned fibres, as already reported from other techniques. While we have done this, we must point out that any of these changes in other structures is very much secondary to the importance of the changes in the t-system structure and its Ca^{2+} -handling ability that we have demonstrated here.

-The authors mention that differences in fiber types might influence the results. It seems that this is an important issue given that Ca^{2+} handling differs between fiber types. It seems knowing whether vacuoles are formed in slow v fast or vice versa is important to this phenomenon.

We now report in the revised ms that vacuoles occur regardless of fibre type. We also note in the revision that Yeung et al 2002 and Warren et al 1993 show the presence of vacuoles following post-eccentric contractions in both fast- and slow-twitch fibres of mouse. (see top of p. 6, Results).

-What is the relationship of the mitochondria and vacuoles? Is the Ca handling energy requiring?

The vacuoles require the Na^{+} -pump to be functional to persist (Yeung et al 2002; Sim & Fraser, 2014). Therefore we can conclude that mitochondria must be functional in the intact muscle to maintain the Na^{+} -pump and therefore the vacuoles; and, it follows, to maintain the Ca^{2+} that the vacuoles sequester. Consistent with this, Nielsen et al 2015 *PLoS ONE* have described an increase in glycogen metabolism in the days post-eccentric exercise that may in part be the consequence of the support of the energy requirement for storage of Ca^{2+} in vacuoles. This has been added to the revised ms (see p.10, para 3, Discussion).

-Since the vacuoles are rather large, can they be isolated (differential centrifugation) using standard biochemical techniques. This would enable a more thorough characterization of the vacuoles offering important validation of the imaging studies. Are there differences in the expression of proteins b/n the vacuoles and TT?

Firstly, it is important to point here that vacuoles are transient structures that require the persistent presence of ATP and intrinsic hydrostatic pressure. It is not feasible to maintain such conditions throughout a fractionation process.

Additionally, there are many assumptions in experiments where fractionation of membranes from cells has been performed. There is no biochemical or biophysical evidence to suggest that the vacuoles could fractionate by size, without rupture, during homogenization procedure. Noting the lack of evidence that there may be a redistribution of the phospholipids between transverse and vacuolating longitudinal tubules, it can only be expected that as these structures consist of extreme intrinsic hydrostatic pressure which would allow it to rupture rather than reseal in its original size. We also point out that there are no previously-identified biomarkers which could differentially confirm the biochemical identity of vacuoles (or longitudinal tubules) as a crucial validation step before we carry out a biochemical analysis of Ca handling proteins such as STIM1, PMCA and NCX. Further, our *in situ* imaging measurements estimate the longitudinal tubules (which undergo vacuolation) to consist of a mere 1% of the t-system membranes (derived from the vol-tsyst measurement $\sim 1\%$). In the absence of any biochemical references and the relatively high variability associated with antibody detection efficiency ($\sim 10\%$, our estimate), an *in vitro* differentiation of

vacuolar membrane fractions from any contaminants (e.g. transverse tubule membranes, surface sarcolemma in intact fibres) is not feasible.

More physiological preparations should always be preferred where such a preparation has been developed. In the manuscript we have advanced the skinned muscle fibre preparation so that Ca handling by vacuoles can be quantitatively examined and we do not need to resort to fractionation. In situ imaging of longitudinal tubules hold a clear advantage over this type of biochemical approach because of the precision in accurately resolving these structures. Whilst we appreciate that it would be useful in the future to develop biochemical methods to characterise these membranes we respectfully argue that the imaging tools, because of the extensive image data spanning the last 2 decades, should be the benchmark for a validation measure, not vice versa.

Our imaging studies of the Ca²⁺-handling of the t-system have been well validated through a series of papers (Launikonis et al 2003, PNAS; Launikonis & Rios, 2007, J.Physiol; Edwards et al 2010, Cell Calcium, 2010, AJP, 2011, Aging Cell; Cully et al 2016, J.Physiol). We do not believe any further validation of our experimental technique is required.

Reviewer #2, expert in muscle physiology
Has submitted his/her review as a PDF.

In skeletal muscle the tight control of cytosolic calcium is critical to maintain proper cellular function. When cytosolic calcium is in excess, it drives oxidative stress and calpain proteases, which disrupt signaling pathways and EC coupling structures.

While most in the field have focused the mechanisms for calcium associated muscle dysfunction in disease, we still have critical gaps in our understanding of how calcium is managed in the healthy skeletal muscle cell; especially during and after severe fatiguing exercise or injurious muscle contractions. Revealing these mechanisms in healthy muscle is critical to *fully understand* why calcium handling is dysregulated in disease.

This manuscript by Culley et. al. offers timely and intriguing evidence that the functional remodeling of the t-tubular system is a central player in how the skeletal muscle manages cytosolic $[Ca^{2+}]$ after a bout of un-accustomed exercise. Specifically, these authors demonstrate that the longitudinal t-tubule system is a privileged participant in this process by forming vacuoles that sequester Ca^{2+} . Support is provided using a host of advanced imaging techniques to quantify Ttubular morphology and function. The use of biopsy material from human participants pre- and post-exercise is a powerful approach to address this question. While this reviewer finds significant merit in this work, there are several areas in this manuscript that may diminish its impact and accessibility to the field.

We wish to thank the reviewer for their time and positive comments on our manuscript.

Major Comments:

1. Taken together the results shown support a model in which the vacuolization of the longitudinal t-system (LT) is a 'saftey net' by which the muscle cell can sequester excess myoplasmic Ca^{2+} associated with exhaustive exercise. If successful, the muscle cell would be able to maintain it viability while it recovers from this insult.

a. "If" this is an accurate summary of the major results, the introduction should be re-tooled to better set-up this concept. For example, the statement; Line 93: "*A compartmentalization of Ca^{2+} in vacuoles would be an advantage in the days following heavy-load resistance exercise, when delayed onset muscle soreness and low-frequency or long-duration fatigue are experienced (24).*" These vacuoles would have protected the muscle such that 'fatigue' was not irrecoverable. Rather it is reversible with muscle soreness and long duration fatigue being a consequence.

b. Line 86: "*Vacuole formation has been associated with fatigue and lactate movements in mouse muscle. (17, 18) but the physiological importance of vacuolar accumulation of small molecules or ions following exercise has not been explored.*" If vacuole formation has been studied in fatigue, it has been studied during 'exercise'. Please rephrase.

a. Line 96: The sentence "*To provide evidence for a role of the t-system as a dynamic Ca^{2+} buffer.....*" Suggests that this experiment was not hypothesis driven, rather an experiment to confirm an established

phenomenon. As I trust this is not the author's intention, please revise.

We thank the reviewer for highlighting a better way for presenting our results. In the revised manuscript, we have completely re-written the Introduction, and Abstract and revised the Discussion sections encompassing the above points to clarify the hypothesis and the aims of this body of research.

2. Line 177: The authors state: "*regions abundant of longitudinal tubules were identified; often these extended as a series of longitudinal tubule networks across multiple sarcomeres (Fig 1B), typically in between myofibrils that lack alignment between sarcomeres (suppl Fig S1 E&F)*

a. Can the authors elaborate on the finding that there are 'regions' of these LT's?. If these are seen in heterogeneous localization, this is likely not a physiologically relevant structure at baseline and rather a consequence of an adaptation.

We and others have reported structural indicators of sarcomere misalignment in toad and rat skeletal muscle and rat cardiomyocytes previously (Peachey & Eisenberg, 1978, *Biophys J*; Launikonis & Stephenson, 2004, *JGP*; Edwards & Launikonis, 2008, *J.Physiol*; Jayasinghe et al 2010, *JMCC*). Eccentric contraction or indeed any other intervention was not a factor in any of these studies.

Heterogeneity of sarcomere misalignment seems to be the basal condition. We now comment on this possibility in the revision (see p.10, para 1, Discussion).

b. The idea that these LT's occur at areas of sarcomere/myofibrillar misalignment may suggest that they have arisen secondary to a damage and repair consequence (see 1,2).

The current understanding across a large body of research suggests that turnover of myofibrils in the muscle is a normal part of its physiology. However, such misalignment is also present within the healthy cardiomyocytes at both systolic and diastolic levels of stretch (Jayasinghe et al. *JMCC* 2010; Soeller et al *Exp Physiol* 2009). This evidence argues that myofibril misalignment in muscle is by no means purely a consequence of damage repair. The situation in dystrophy is much more severe and there are much more tortuous and twisting interactions between neighbouring myofibrils in dystrophic muscle than in healthy muscle (our own unpublished comparisons of single fibres of WT and mdx muscle; Buttgerit et al 2013).

We also point out that the t-system navigates around local structures such as nuclei that are between myofibrils and not uniform throughout the fibre (Edwards & Launikonis, 2008, *J Physiol*; Jayasinghe & Launikonis, 2013, *J Cell Sci*). We can expect non-uniform distribution of longitudinal tubules for a number of reasons.

The above points are now raised in The Discussion, see p.10, para 1.

This can occur due to physical damage or activation of calpains that cleave desmin which hold these myofibrillar structures together. In fact, there is an intriguing hypothesis that this desmin cleavage may also be a 'safety net' that prevents damage with

exhaustive exercise (see 3-5).

We see that the loss of desmin may help reduce the eccentric contraction induced damage to a muscle. We now include mention of desmin in the revision (see p.9, para 3, Discussion).

c. The authors state that the biopsies were collected in the basal condition 'prior to exercise'.

i. Which group was used to support the statement in Line 177? The recreational athlete? The trained exercisers?

This biopsy was from a recreationally active individual. No training or heavy exercise was performed by this individual in the 2-3 days prior to biopsy. We have revised the Methods section to make this point more explicit (see p. 11, Methods, p. 12, para 1.).

How long prior to the last bout of exercise was the biopsy taken?

Acute biopsies were taken at '0' (~30 min before exercise), and then 24 and 48 hours after exercise. In the longitudinal study, the '0' biopsy was collected 4-5 days before the first exercise training session, whilst the 'post-exercise' biopsy was collected 6-7 days after the last training session. This information is now provided in the supplemental Methods.

ii. It may be important to comment on the basal morphology of both groups (novice and trained) as differences in their occurrence may provide insight.

We examined a high spatial resolution reconstruction of the t-system from a recreationally active individual. His LT % was similar to that in the training experienced men. However, there are a number of issues comparing these groups because the "trained" athletes were amateurs and our "novices" were recreationally active, thus the groups are not highly distinct. We prefer not to comment on this aspect, although there may be a difference between professional athletes and others, we have not done such a study to determine this. We wish to avoid such statements at the moment.

iii. Did the authors note more sarcomere miss-alignments along with the vacuolization in the muscle fibers after exhaustive exercise?

There is a heterogeneous change in sarcomere alignment after eccentric exercise (Friden et al 1981, *Experientia*). However it is difficult to quantify whether there are more sarcomeres misaligned along with vacuolation because our reference, the transverse tubules, become much less transversal in the post-exercised muscle (Fig 2E). For this reason we do not comment on this in the revision. See p.10, para 2, Discussion.

3. The authors are obliged to provide more detail regarding the origination of the muscle samples.

a. These authors need to state implicitly that this current study used samples from a recently published study. They then need to provide details on the subjects and their exercise regimes. If its duplicative, put it in supplemental methods as is not appropriate to reference a published 2015 paper in which muscle biopsies were taken from multiple groups and treatment conditions.

We have provided the full detail of the training regimes to induce DOMS in a new Methods section in the supplement.

b. In fact this is potentially perilous to piggy back on this 2015 study as they

focused on a 'rescue' condition (cold treatment) that should reduce the amount of vacuoles if these authors hypothesis is true. Do these authors have this data?

Our results for 0, 24, 48 hrs after heavy load resistance exercise were with 2 individuals receiving cold treatment and one without. There was no difference noticed between the treatments. For this reason the data has been pooled. 3 of 6 subjects who embarked on the 12-week training regime were exposed to cold water immersion. No obvious difference in t-system structure between the 2 groups was observed. This is now stated in the supplemental Methods. The expected effect of cold treatment on vacuoles would be, if anything, to slow the action of the Na⁺ pump, which may briefly reduce vacuoles. Yeung et al 2002 show that eccentric contraction that was followed by ouabain (Na⁺K⁺-ATPase inhibitor), inhibited the formation of vacuoles until after ouabain had been washed-out.

4. Line 285: The authors state *“The ‘trapping’ of the Ca²⁺ within the vacuoles in the presence of chronically activated SOCE, may be assisted by funnel-like collars at the luminal junction between the transverse tubules and vacuoles (9). Such structures would maintain a pathway for the entry of Ca²⁺ into the vacuoles from the transverse tubules, but make the exit of Ca²⁺ through a very small aperture an extremely slow process.”*

a. The authors appear to be taking liberties with the structures they are defining. In the 2008 paper (ref 9) the authors were describing the movement of ions in the LT and suggesting they moves slower based on restricted space. Here the authors suggest that vacuoles are cut-off from the rest of the t-system. If that is the case, there should be no tunneling into a vacuole, rather there should be active transport into this vacuole.

We take this point and have revised the manuscript accordingly. We state that the connection between the lumen of the vacuoles and transverse tubules is significantly restricted. This is line with the loading of dye into the vacuoles via diffusion from the transverse tubules, shown in our previous study (Edwards & Launikonis, 2008, *J. Physiol*). See p.10, para 3, Discussion.

i. Is this the case? If so, can the authors comment on the means by which the calcium is transported in?

The calcium is most likely transported into the vacuole?? via NCX and/or PMCA which have been identified from vesicle studies (Largely the work of Cecilia Hidalgo). We have cited these studies in the revision. (see p. 10, para 3, Discussion).

ii. How does the calcium accululate to such a great extent in the vacuole? Is their trapped buffer inside the vacuole to allow calcium to accumulate to the great extent that is predicted?

The vacuole has a large volume relative to the volume of transverse tubules, allowing it to accumulate ions and molecules, as we have demonstrated previously (Edwards & Launikonis, 2008). It is the increase in volume that we have used to calculate a minimum of calcium that accumulates in the vacuoles (Table 1). It is possible diffusible buffers also enter the vacuoles, to increase the capacity to hold calcium. Additionally, the vacuoles may be depolarized, which would also provide an environment where the movement of calcium would be electrochemically favoured to enter. These issues are now raised in the revision. See p.8, Results.

b. Can the authors speculate as to why this vacuolization occurs primarily near the muscle fiber surface? And not throughout the myofiber?

The reason for this is the vacuoles form from longitudinal tubules at sarcomere mis-alignments (Edwards & Launikonis, 2008) and as pointed out above as well, the mis-alignments form mostly near the fibre periphery (this study). See p.10, para 1, Discussion.

REVIEWERS' COMMENTS:

Reviewer #1 (Remarks to the Author):

In this manuscript the authors identify remodeling events of TT system in response to eccentric exercise. These changes involve formation of transverse oriented TT and eventual vacuole formation that may function as Ca²⁺ sink. The authors have been pretty responsive to the reviews. The writing is much clearer. The finding that eccentric exercise remodels that TT is an important one but some issues remain. Limitations of the study include the fragmented analysis for TT remodeling and Ca²⁺ imaging. It appears these studies were done on separate groups of subjects. There is also little mechanism provided to understand how these remodeling events take place. Calpains are proposed and discussed(as suggested by the authors)--could this not be tested.

Reviewer 2 raised issues related to the use of samples from the 2015 study. I also expressed reservations about the low numbers of subjects. The authors' state that the first study involved three subjects with and without a treatment (cold immersion)and the data were pooled. This raises a lot of issues. For example why did not the cold treatment limit or alter the vacuole formation. Along the same lines the Ca²⁺ studies are done from 7 subjects. While these Ca²⁺ studies clearly support the hypothesis and represent an important advance, I remain concerned about the overstatement of the relationship between fiber type and other correlative data.

Response to reviewer's comments.

REVIEWERS' COMMENTS:

Reviewer #1 (Remarks to the Author):

We wish to thank the reviewer for their time.

In this manuscript the authors identify remodeling events of TT system in response to eccentric exercise. These changes involve formation of transverse oriented TT and eventual vacuole formation that may function as Ca²⁺ sink. The authors have been pretty responsive to the reviews. The writing is much clearer. The finding that eccentric exercise remodels that TT is an important one but some issues remain. Limitations of the study include the fragmented analysis for TT remodeling and Ca²⁺ imaging. It appears these studies were done on separate groups of subjects. There is also little mechanism provided to understand how these remodeling events take place. Calpains are proposed and discussed (as suggested by the authors)--could this not be tested.

The TT remodelling and Ca²⁺ imaging experiments have been performed separately for several reasons. Firstly both types of measurements require a significant amount of time to acquire the images, which must occur within the limited amount of time that the biopsy remains viable for such work (typically up to 12 hrs post-biopsy). The exercise study was difficult to perform because it required subjects willing to be biopsied over consecutive days, in addition to the logistical issues associated with this. It was possible to acutely induce vacuoles in non-exercised biopsy, so Ca²⁺ imaging experiments could be performed in a less strict manner, without compromising the integrity of the results. The Ca²⁺-handling differences between transverse tubules and vacuoles are expected to be the same in non-exercised and exercised individuals. In contrast, the change in t-system structure with exercise was fundamental, and had to be imaged following exercise.

The mechanism of vacuolation has been described in some detail by Yeung et al 2002 and Sim & Fraser, 2014 (both papers cited in revision). An increase in cytoplasmic Na⁺ and/or Ca²⁺ in the cytoplasm is the trigger for the activation of the Na⁺ pump. Na⁺ is pumped into the longitudinal tubules, followed by water, to cause the increase in volume.

The role of calpains in eccentric exercise is also discussed (Raastad et al 2010).

In addition we must point out that some of the previous suggestions of the reviewer to 'tease-out' mechanism here were not suitable for application in biopsied human muscle. Our approach working with human muscle is highly novel and outweighs moving to cell culture approaches or similar where conditions can be manipulated to a greater extent (but most likely lose physiological context, which we maintain).

Reviewer 2 raised issues related to the use of samples from the 2015 study. I also expressed reservations about the low numbers of subjects. The authors' state that the first study involved three subjects with and without a treatment (cold immersion) and the data were pooled. This raises a lot of issues. For example why did not the cold treatment limit or alter the vacuole formation. Along the same lines the Ca²⁺ studies are done from 7 subjects. While these Ca²⁺ studies clearly

support the hypothesis and represent an important advance, I remain concerned about the overstatement of the relationship between fiber type and other correlative data.

There are a high number of repeat measures performed on each biopsy in the study by using many fibres from each biopsy for the same measurement. For the two major sections of the study, the t-system structural analysis following exercise used three subjects, and 24 fibres. The n value is 24, not 3. The structural changes in the t-system post-exercise are highly significant, as presented in Fig 2. Additionally, the calcium imaging section of the study, were results obtained from 27 fibres (from 7 people), with multiple values obtained from single fibres in many cases (this is listed at end of Fig 3 legend). Sampling needs to be viewed in the context of how many fibres were have imaged, which is more than adequate, not the number of biopsies.

Additionally, changes in t-system structure were not observed in those 7 individuals used for the calcium section (unless induced by acute application of high Ca^{2+}). Therefore the number of subjects consistent with producing a significant change in t-system structure (i.e. inducing vacuoles) post-heavy exercise is 10.

The cold water immersion in the 2015 study was only 10 min! The mechanism of vacuolation post-eccentric exercise requires the function of the Na^+ pump (Yeung et al 2002, which we cite). The Na^+ pump may be slowed by lowered muscle temperature. Any slowing would only last while muscle temperature was indeed lowered. The important point is that the resolution of our measurements of t-system structural changes was 24 hrs. Biopsies were taken at 0, 24 & 48 hrs from the exercises. Even if there was *any* change in vacuole rate of formation, it could only be in the order of tens of minutes. Such a delay is not resolvable with our measurements. The cold water immersion is clearly not significant; and we would suggest it is not reasonable to expect or suggest that a significant change in the t-system biology we have described would result from such brief exposure to cold water.

Reviewer 1: "I remain concerned about the overstatement of the relationship between fiber type and other correlative data"

We don't assign physiological characteristics based on fibre types in this work, just identify the fibre types from each fibre assayed in our presentation of results (Fig 3); and also report that all fibres imaged to post-heavy exercise vacuolated (19/19 fibres), which we would expect therefore, to occur across the fibre types. There is no overstatement.